

# Effect of water-based aerobic training on anthropometric, biochemical, cardiovascular, and explosive strength parameters in young overweight and obese women: a randomized controlled trial

Imen Ben Cheikh[1,2], Hamza Marzouki[1,2], Okba Selmi[1,2], Bilel Cherni[1,2], Siwar Bouray[1,2], Ezdine Bouhlel[3], Anissa Bouassida[1,2], Beat Knechtle[4,5] and Yung-Sheng Chen[6,7,8]

[1] Research Unit: Sport Sciences, Health and Movement, University of Jendouba, El Kef, Tunisia
[2] High Institute of Sport and Physical Education of Kef, University of Jendouba, El Kef, Tunisia
[3] Laboratory of Cardio-Circulatory, Respiratory, Metabolic and Hormonal Adaptations to Muscular Exercise, Faculty of Medicine Ibn El Jazzar, University of Sousse, Sousse, Tunisia
[4] Institute of Primary Care, University of Zurich, Zurich, Switzerland
[5] Medbase St, Gallen Am Vadianplatz, St. Gallen, Switzerland
[6] Exercise and Health Promotion Association, New Taipei City, Taiwan
[7] High Performance Unit, Chinese Taipei Football Association, New Taipei City, Taiwan
[8] Department of Exercise and Health Sciences, University of Taipei, Taipei, Taiwan

Corresponding author
Yung-Sheng Chen,
yschen@utaipei.edu.tw

## ABSTRACT

**Background.** Obesity is a major health concern that raises the risk of chronic illnesses such as heart disease, diabetes, and metabolic disorders. Traditional workouts such as running or walking can be difficult for overweight individuals due to the heavy impact on joints, which causes discomfort and the possibility of injury. Water-based exercises offer a low-impact alternative that overweight people may find more tolerable. There is minimal research on the specific effects of structured water aerobic exercise on health markers in young overweight and obese women, despite the acknowledged benefits of physical activity for weight control and overall health. This study aimed to assess the effects of 10-week water-based aerobic training (thrice a week) on anthropometric, biochemical, cardiovascular parameters, and explosive strength in young overweight and obese women.

**Methods.** In a randomized controlled trial with a pre-to-post testing design, twenty-seven young overweight and obese women (age: $27 \pm 1$ years; body mass index (BMI) = $30.0 \pm 3.1$) were randomly assigned into experimental (EG: performing a water-based aerobic training, $n = 16$) or control (CG: maintaining their usual activities during the intervention, $n = 11$) groups. The pre- and post-intervention participants were assessed for their anthropometrics (body height, body weight, body mass index (BMI), body fat and circumferences), biochemical (fasting glycemia, total cholesterol, high-density lipoprotein cholesterol (HDL-C), low-density lipoprotein cholesterol (LDL-C) and triglyceride (TG)), cardiovascular parameters (resting blood pressure and resting heart rate (RHR)), and explosive strength of upper and lower limbs.
**Results**. EG showed reductions in body weight, BMI, %BF, fasting glycemia, and TG, along with improvements in HDL-C, LDL-C, RHR, and explosive strength (all $p < 0.05$; effect size (ES) = 0.180–1.512, trivial to large). In contrast, CG exhibited increases in body weight, BMI, fasting glycemia, LDL-C, and RHR (all $p < 0.05$; ES = 0.127–0.993, trivial to large), with no significant changes observed in other measured variables. EG showed superior post-test results in fasting glycemia ($p < 0.0001$; ES = 2.559, large), LDL-C ($p < 0.0001$; ES = 0.971, large), and explosive strength measures ($0.003 \leq p < 0.0001$; ES = 1.145–1.311, large) compared to the CG.
**Conclusions**. Our findings indicate that water-based aerobic training could be a useful program to enhance anthropometric, biochemical, cardiovascular, and explosive strength parameters in young overweight and obese women compared to inactive persons.

# INTRODUCTION

Obesity is a serious worldwide health problem associated with increased risks of cardiovascular diseases, diabetes, and musculoskeletal and metabolic disorders (*Welsh et al., 2024*; *Piche et al., 2018*; *Neeland, Poirier & Despres, 2018*). The prevalence of obesity continues to rise globally, making it one of the leading public health challenges. Fortunately, physical activity is a key factor in the prevention and treatment of obesity (*Welsh et al., 2024*; *Oppert et al., 2021*). Regular physical exercise is well-documented to have a positive effect on physical performance, and it is associated with numerous beneficial changes in anthropometrics and health markers, such as weight loss, improvement of glycemic control, changes in lipid profile, and gains in muscle strength and oxygen uptake among both overweight and obese adults (*Welsh et al., 2024*; *Pippi et al., 2022*; *Costa et al., 2018*; *Delevatti, Marson & Kruel, 2015*). However, traditional modalities of exercise (*i.e.,* walking, running) are linked to a higher risk of musculoskeletal injury because they impose an increased load on the lower limbs (*Pippi et al., 2022*). This risk is particularly high in obese individuals, as their excess body weight can lead to significant stress and pain in the joints of the lower limbs, thereby discouraging physical activity (*Pippi et al., 2022*; *Zdziarski, Wasser & Vincent, 2015*). Even though the health advantages of aerobic exercise training have been demonstrated (*Pippi et al., 2022*), the challenge of joint discomfort and injury risk remains a significant barrier.

Water-based exercises, such as swimming, water aerobics, and deep-water running, offer a low-impact alternative that is often more tolerable for overweight and obese individuals than land-based exercises (*Torres-Ronda & Del Alcázar, 2014*; *Donnelly et al., 2009*). The buoyancy provided by water reduces the impact on weight-bearing joints, thus alleviating the mechanical stress that can lead to pain and injury during traditional exercises. Moreover, the resistance of water provides a natural form of resistance training, enhancing muscle strength and endurance without the need for additional equipment (*Torres-Ronda &*

*Del Alcázar, 2014*). These properties make water-based exercises particularly suitable for overweight and obese individuals, who may find it challenging to engage in high-impact land-based activities due to their body weight (*Pippi et al., 2022*).

Numerous studies have investigated the effect of water training on physical health and have shown a wide range of benefits for different populations, including healthy individuals (*Pereira Neiva et al., 2018*; *Kantyka et al., 2015*), persons with cardiovascular diseases (*Scheer et al., 2021*; *Volaklis, Spassis & Tokmakidis, 2007*), diabetes (*Delevatti et al., 2016*; *Scheer et al., 2020*), and obese persons (*Bielec, Kwasna & Gaworska, 2017*). Additionally, water-based exercise allows for greater adherence to physical activity programs, as the enjoyable and less painful nature of the activity increases the likelihood that participants will continue to engage in regular exercise (*Ambrose & Golightly, 2015*). Specifically, training programs with aerobic features are known to promote health status in obese women (*Nosrani et al., 2023*; *Bielec, Kwasna & Gaworska, 2017*; *Abadi et al., 2017*; *Penaforte et al., 2015*). According to *Bielec, Kwasna & Gaworska (2017)*, six months of aqua-power aerobic training led to a significant decrease in anthropometric parameters (*e.g.*, body mass, body fat, body mass index—BMI) and heart rate values in middle-aged obese women (36 to 57 years). Similarly, *Abadi et al. (2017)* demonstrated that 12 weeks of aqua aerobics improved cardiovascular fitness and induced weight loss in young female students. *Penaforte et al. (2015)* found that two months of water aerobics had little effect on anthropometrics and metabolic profile in obese women (~42.8 years).

The existing literature shows significant variations in participant age, exercise intensities, program durations, and objectives, which complicates the comparison of results and raises questions about the actual effects of water-based training. Most protocols employ limited tasks and/or are set up by researchers with a lack of investigation into real-life settings. Therefore, further information is warranted to comprehend the impact of adequate aquatic programs in genuine scenarios on improving obese women's health. Moreover, the effects of short-term water-based aerobic programs using interval training at moderate to high intensities on health risk factors (*e.g.*, weight loss, BMI, biochemical and cardiovascular parameters) and physical fitness in inactive young obese women remain uncertain. The choice of assessing anthropometric, biochemical, cardiovascular, and explosive strength parameters is rooted in their importance as indicators of overall health and physical fitness. Improvements in these parameters are directly related to reduced health risks and enhanced quality of life for overweight and obese women (*Welsh et al., 2024*; *Oppert et al., 2021*). Specifically, explosive strength is critical for daily activities and overall physical performance, making it a valuable measure in evaluating the effectiveness of exercise interventions (*Newton & Kraemer, 1994*). Therefore, this study aimed to assess the effects of a 10-week water-based aerobic training program without nutritional intervention (thrice a week) on anthropometric, biochemical, cardiovascular, and explosive strength parameters in young overweight and obese women. We hypothesized that a 10-week water-based aerobic training program would lead to statistically significant improvements across various health metrics in young overweight and obese women. Specifically, we anticipate reductions in anthropometric measures, enhancements in biochemical markers,

**Table 1** Participants' characteristics at baseline for the experimental (EG; $n = 16$) and control (CG; $n = 11$) groups.

| Groups | Age (years) | Height (cm) | Weight (kg) | BMI (kg m$^{-2}$) | BF (%) |
|---|---|---|---|---|---|
| EG | $26.4 \pm 4.1^{\dagger}$ | $165.0 \pm 0.1^{\dagger}$ | $81.2 \pm 8.2$ | $29.9 \pm 3.1$ | $37.4 \pm 4.0$ |
| CG | $23.3 \pm 1.4$ | $161.0 \pm 0.0$ | $78.3 \pm 8.1$ | $30.1 \pm 3.1$ | $36.4 \pm 3.3$ |

**Notes.**

Values are given as means $\pm$ standard deviation.

BMI, body mass index; BF, body fat.

$^{\dagger}$Significantly different from CG at pre-test.

The statistical significance level was set at $p \leq 0.05$.

and cardiovascular health improvements. Finally, we hypothesized an increase in explosive strength in both the upper and lower limbs.

## MATERIALS & METHODS

### Participants

Fliers and advertisements about the study were used to recruit the participants who work in regional academic establishments, live in the same city (Kef, Tunisia), and from a similar socio-economic status. The criteria for participating in the study included: (a) being in a stable state of health, specifically not having any infectious or skin diseases; (b) being of young age (*ACSM, 2018*); (c) having BMI indicating overweight (BMI ranged between 25.0 and 29.9 kg m$^{-2}$) and obese (BMI ranged between 30.0 and 34.9 kg m$^{-2}$) status (*World Health Organization, 2000*); and (d) having prior experience with water-based exercise routines. Thus, an initial sample of 37 women agreed to take part in the study. Participants were informed about the purpose of the study and their possible risks and benefits before the start of the study. History of familial or individual cardiovascular disease and treatments were collected for each participant. As mentioned above, subjects with a history of cardiovascular disease, hypercholesterolemia, diabetes, arterial hypertension, orthopedic limitations, water phobia or any other contraindication for aquatic exercise, and inability to safely enter and exit the pool were excluded. To ensure concealment of allocation, eligibility was determined by a blinded assessor not involved in the randomization process. Thirty-one subjects met the criteria and were diagnosed as overweight or obese according to BMI (BMI > 25 kg m$^{-2}$) (Table 1). None of the subjects were using drugs or other therapy for obesity, and all were sedentary over the previous six month.

Random allocation was maintained using the method of randomly permuted blocks matched by BMI, which resulted in the following assignments: experimental group (EG; performing a water-based aerobic training program, $n = 16$) and control group (CG; maintaining their usual activities for 10 weeks, $n = 15$). In addition, all participants were asked not to make any changes to their daily diet and not to engage in any dietary practices that would result in weight loss. To be included in the final analysis, the participants should complete at least 90% of the total training sessions and the testing sessions, which resulted in excluding the data of four CG participants from the final analyses. Figure 1 displays the consolidated standards of reporting trials (CONSORT) flow chart of the study (*Moher et al., 2010*). The demographic and anthropometric characteristics of both groups are illustrated in Table 1. This study received institutional ethics approval (Research Ethics

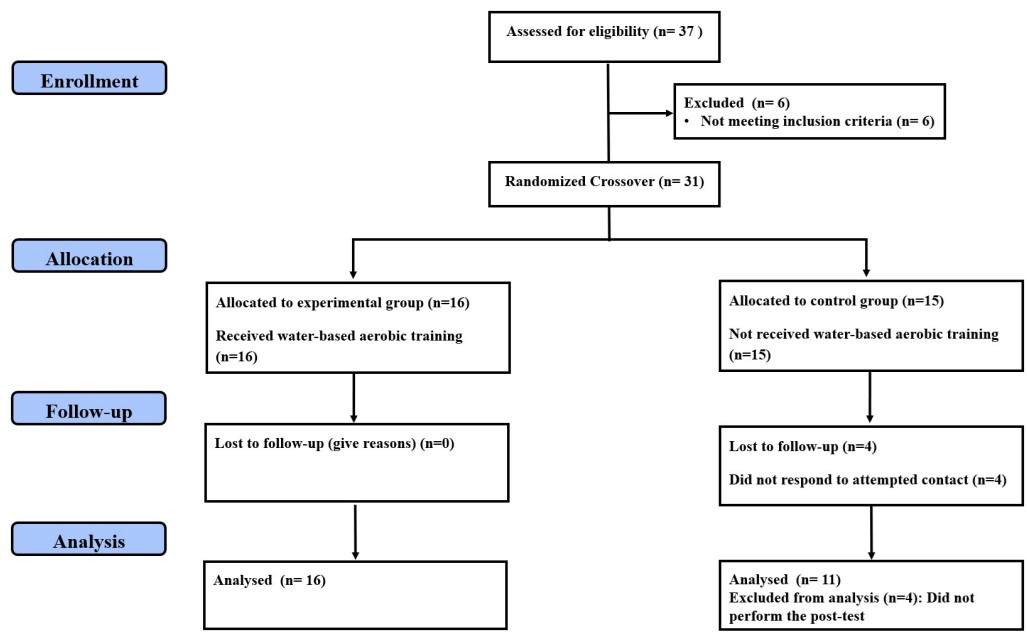

**Figure 1  CONSORT diagram of participants' recruitment, allocation, follow-up and analysis.**

Committee of the High Institute of Sport and Physical Education of Kef, University of Jendouba, approval number: 014/2018; date of approval: October 17, 2018), and volunteers provided written, informed consent forms before starting their participation in the trial. This study was registered in ClinicalTrials.gov with a code of NCT06371105.

## Procedures

The present investigation consists of a randomized controlled trial with a pre-to-post testing design to examine the changes in anthropometrics, lipid profile, cardiovascular and explosive strength parameters of young adult overweight/obese women after 10 weeks of water-based aerobic training. Participants were assigned into EG or CG. Overall, the study lasted 12 weeks, comprising 1 week for pre-testing, 10 weeks dedicated to the training intervention, and 1 week for post-testing. It was conducted at the High Institute of Sport and Physical Education in Kef, Tunisia, from January to March 2019. One week before starting the experimentation, all participants completed three separate familiarization sessions with the experimental procedures (~48 h) to ensure their technical proficiency in performing testing and training procedures, and with the Borg Scale of Perceived Exertion 6–20 (RPE) (*Borg, 1982*). The results of the familiarization and pre-test sessions were used for assessing the relative and absolute reliability (*i.e.*, intraclass correlation coefficient (ICC) and standard error measurement (SEM)) of the explosive strength measures. The following variables were evaluated the week before the training implementation (*i.e.,* pre-test) and the week after the end of the program (*i.e.,* post-test): anthropometrics (body height, body weight, BMI, body fat and circumferences), biochemicals (fasting glycemia, total cholesterol (TC), high-density lipoprotein cholesterol (HDL-C), low-density lipoprotein cholesterol

(LDL-C) and triglyceride (TG)), cardiovascular parameters (resting blood pressure and resting heart rate—RHR), and explosive strength (seated medicine ball throw test (MBT), countermovement jump (CMJ) and squat jump (SJ)).

All variables were assessed and overseen by the same investigators who were not blinded to groups, and testing was conducted on the same equipment, with identical participant/equipment positioning, at the same time of day over two non-consecutive days 48 h apart (Monday and Wednesday). The morning of the first day was devoted to anthropometric, biochemical, and cardiovascular measures. On the evening of the second day (19:00–21:00), the participants performed the explosive strength tests in the indoor basketball court of the High Institute of Sport and Physical Education (temperature: 19–25 °C, resilient flooring) under verbal encouragement. Each explosive strength test involved three valid maximal trials interspersed by 2 min of passive recovery, and the best performance was retained for analysis. Five minutes of passive recovery were allowed between tests. Before the testing session, the participants performed a 15-min warm-up including jogging, stretching, and intense exercises (*e.g.*, short sprints and skipping). During the evaluation period, the participants were instructed to refrain from physical activities, and caffeine and alcohol consumption.

## Measures
### Anthropometric measures
Body weight (kg) was measured with an accuracy of 0.1 kg using a digital scale (OHAUS, Florhman Park, NJ, USA), body height (m) was recorded with a stadiometer to the nearest 0.1 cm (Seca model 213), and BMI was determined using the standard formula: BMI $(\text{kg m}^{-2}) = \text{body weight (kg)} \times \text{body height}^{-2} \text{ (m)}$. A standard caliper (Harpenden/Holtain Calipers; Crosswell, Crymych, Pembrokeshire, UK) was used to measure triceps, suprailiac and thigh skinfolds (*American College of Sports Medicine, ACSM)(2018)*). Measures were collected and used to estimate the body density (*Jackson, Pollock & Ward, 1980*) and the body fat percentage (%BF) based on the equation of *Siri (1961)*. All measurements were made on the left side of the body and participants were asked to use the least amount of clothing possible.

### Biochemical assessment
As a routine procedure, four ml blood samples were collected from the antecubital vein of each participant after ∼12 h of overnight fast. The samples were centrifuged at 3,000 rpm for 15 min to obtain blood serum which was analyzed on the same day. Fasting glycemia, TC, HDL-C, LDL-C, and TG concentrations were analyzed with Indiko kits (Indiko, Thermo Fisher Scientific, Oy, Vantaa, Finland) using the respective automated analyzer (Indiko, Thermo Fisher Scientific, Oy, Vantaa, Finland).

### Blood pressure and heart rate assessment
The resting systolic (RSBP) and diastolic blood pressure (RDBP) levels (mmHg) were collected *via* auscultation using a hand-held sphygmomanometer and stethoscope with the arm resting on a table and parallel to the heart level. HR was recorded continuously with a 5-second interval for 1 min *via* a telemetric system (Polar Team2 Pro System; Polar

Electro OY, Kempele, Finland), and the lowest value was used for analysis. The readings were taken after 10-min rest in a seated position and a noise-free environment.

### Upper limb explosive strength assessment

A seated medicine ball throw test was adopted to assess the upper limb explosive strength following the protocol of *Vossen et al. (2000)*. Participants performed the test from a sitting position and using a three-kilo medicine ball (Dynamax Inc®, Dallas, TX, USA). Each participant was instructed to sit on an adjustable bench with the back in contact with vertical back support, thighs horizontal, knees bent at a 90° angle, and ankles fixed behind swivel pads at the base of the bench. The ball was held with both hands and with forearms positioned parallel to the ground at the height of the sternum. For the throw, the participant was instructed to perform an explosive chest-type pass as far straight forward as maintaining the back against the back support. Distance from the base of the bench to where the ball lands was recorded. The ICC and SEM were 0.887 and 4.44%, respectively.

### Lower limb explosive strength assessment

The SJ test was adopted to measure the explosive lower-body strength (height in cm) following the protocol of *Bosco, Luhtanen & Komi (1983)*. Participants performed the test by starting with their knees bent at a 90° angle, and jumping vertically as high as possible with extended knees and ankles, and landing where they took off. Hands should be held on the hips to avoid the effect of arm swinging to the test.

Participants performed a CMJ without an arm swing as described by *Bosco, Luhtanen & Komi (1983)*. From a standing position with the feet shoulder-width apart and the hands placed at the waist, the subjects performed a quick countermovement with the lower limbs followed immediately by a powerful upward motion to jump as high as possible with extended knees and ankles. The jumping height (cm) was measured by using an infrared jump system (Optojump; Microgate, Bolzano, Italy). The ICCs and SEMs were 0.992 and 2.95% for SJ, and 0.980 and 9.79% for CMJ.

## Training program

The water-based aerobic program was adopted from the protocol previously used by *Costa et al. (2018)* and *Costa et al. (2020)*. This training program was carried out thrice a week over 2 non-consecutive days 48 h apart (Monday, Wednesday, and Friday from 18:30 to 19:30 h) for 10 weeks according to ACSM guidelines (*Pescatello et al., 2014*). Water activities were implemented for 50 min which included warm-up (10 min), main activities (30 min), and cool-down (10 min). In each session, participants began with a warm-up that included static walking, a combination of stretching exercises, and water walking for a range of motion and relaxation. The main part adopted an interval training method and consisted of six sets of 3-min effort interspersed by 2 min of active interval at RPE 9 (lighter intensity interval). Participants performed 10 exercises by combining four upper and three lower limb movements. The training load progressively increased during the 10 weeks to minimize the risk of the occurrence of possible injuries in this particular musculature (*Pescatello et al., 2014*). Borg's RPE 6–20 Scale (*Borg, 1982*) was used to decide on the intensity of the program and to assess the progression of all the participants, which was

presumed to be maintained between 12 and 14 RPEs in the first mesocycle (somewhat hard, corresponding to the first ventilatory threshold) and between 15 and 16 RPEs in the second mesocycle (hard, less/around the second ventilatory threshold) (*Alberton et al., 2013a*). The last part would be a cool-down activity (*i.e.,* stretching, deep breathing technique, and relaxation and self-care free water activity). No participants in EG were excluded from the study due to injury. The swimming pool where the intervention took part was 1.50 m deep with water and air temperatures of 30–32 °C and 27–28 °C, respectively. The aquatic activities were instructed by a qualified swimming instructor and under the supervision of an experienced therapist. Participants were asked to track their menstrual cycles throughout the study. Additionally, open communication was maintained with participants about how their menstrual cycles might impact their experience in the study, and flexibility was offered where needed. This helped ensure their comfort and adherence to the intervention. An overview of the program in detail is shown in Table 2. CG participants were instructed to maintain their normal activities and diet throughout the study

## Statistical analysis

Previously to the recruitment procedure, a sample size estimation was conducted using statistical software (G*Power software, version 3.1.9.4; University of Kiel, Kiel, Germany) (*Erdfelder, Faul & Buchner, 1996*). Given the study design (analysis of variance (ANOVA) test with repeated measures, within-between interaction), the effect sizes considered to generate the sample size estimation were attained based on tabled data from previous research (*Faul et al., 2007*). The results established a need for nine participants per group ($f = 0.25$ and actual power $= 80.70\%$) to detect differences with an assumed Type I error of 0.05 and a Type II error rate of 0.20 (statistical power $= 80\%$). All data are expressed as mean $\pm$ standard deviation (SD). The ICC and SEM were calculated to test the reliability of all explosive strength variables. Pre- to-post change percentage ($\Delta$) was calculated for all measures. To verify whether the variables followed a normal distribution, the Shapiro–Wilk test was performed. Moreover, the Levene test was used to check the homogeneity. For all normally distributed variables, a paired $t$-test examined the effectiveness of the training program within both groups. Except for MBT, a two-way analysis of variance (ANOVA) with repeated measures (2 Times $\times$ 2 Conditions) was conducted to examine the main and interaction effects for all variables. Since there is a significant difference at baseline, a two-way analysis of covariance (ANCOVA) with pre-test as a covariate was applied to examine the differences between the groups for CMJ and LDL-C. For the post hoc analyses, the Bonferroni or Games–Howell test was used, depending on the homogeneity of the variances. For MBT (normality not assumed), the Wilcoxon signed-rank test was applied for changes within the same group, while the Mann–Whitney U test was used for comparisons between distinct groups. The effect size (ES) was used to estimate the strength of significant findings, with partial eta squared values converted to Cohen's d ($<0.20$: trivial effect, 0.20 to $<0.50$: small effect, 0.50 to $<0.80$: medium effect, and $\geq 0.80$: large effect) (*Cohen, 1988*). The statistical analyses were conducted using SPSS version 26 for Windows (IBM Corp, Armonk, NY, USA) and the significance level was established at $p \leq 0.05$.
**Table 2  Water-based aerobic program.**

| Weeks | Session | (Sets)×Exercises | Work duration/ Intensity (each exercise) | Rest duration/ Intensity (between sets & exercises) | Total Time (per session) |
|---|---|---|---|---|---|
| Week 1 | Session 1 | (2)×Simultaneous hip adduction and abduction of the 2 legs + abduction and adduction transversal of shoulders, simultaneous of both arms; (2)×Simultaneous knee flexion and extension of the 2 legs + Flexion and extension of elbows with adducted shoulders, simultaneous of both arms; (2)×Simultaneous hip flexion and extension of the 2 legs + Flexion and extension horizontal of shoulders, simultaneous of both arms | | | |
| | Session 2 | (2)×Simultaneous hip flexion and extension of the 2 legs + Flexion and extension horizontal of shoulders, simultaneous of both arms; (2)×Simultaneous knee flexion and extension of the 2 legs + Flexion and extension of elbows with abducted shoulders, simultaneous of both arms; (2)×Simultaneous knee flexion and extension of the 2 legs + Flexion and extension horizontal of shoulders, simultaneous of both arms | | | |
| | Session 3 | (2)×Simultaneous hip adduction and abduction of the 2 legs + abduction and adduction transversal of shoulders, simultaneous of both arms; (2)× Simultaneous knee flexion and extension of the 2 legs + Flexion and extension of elbows with abducted shoulders, simultaneous of both arms; (2)×Simultaneous hip adduction and abduction of the 2 legs + Flexion and extension horizontal of shoulders, simultaneous of both arms | | | |
| Week 2 | Session 1 | (2)×Simultaneous hip flexion and extension of the 2 legs + Flexion and extension horizontal of shoulders, simultaneous of both arms; (2)×Simultaneous hip adduction and abduction of the 2 legs + Flexion and extension horizontal of shoulders, simultaneous of both arms; (2)×Simultaneous knee flexion and extension of the 2 legs + Flexion and extension horizontal of shoulders, simultaneous of both arms | | | |
| | Session 2 | (2)×Simultaneous knee flexion and extension of the 2 legs + Flexion and extension of elbows with adducted shoulders, simultaneous of both arms; (2)×Simultaneous knee flexion and extension of the 2 legs + Flexion and extension of el-bows with abducted shoulders, simultaneous of both arms; (2)×Simultaneous knee flexion and extension of the 2 legs + Flexion and extension horizontal of shoulders, simultaneous of both arms | | | |
| | Session 3 | (2)×Simultaneous hip flexion and extension of the 2 legs + Flexion and extension horizontal of shoulders, simultaneous of both arms; (2)×Simultaneous knee flexion and extension of the 2 legs + Flexion and extension of elbows with abducted shoulders, simultaneous of both arms; (2)×Simultaneous hip adduction and abduction of the 2 legs + Flexion and extension horizontal of shoulders, simultaneous of both arms | | | |
| Week 3 | Session 1 | (2)×Simultaneous hip adduction and abduction of the 2 legs + abbduction and adduction transversal of shoulders, simultaneous of both arms; (2)×Simultaneous hip flexion and extension of the 2 legs + Flexion and extension horizontal of shoulders, simultaneous of both arms; (2)×Simultaneous knee flexion and extension of the 2 legs + Flexion and extension horizontal of shoulders, simultaneous of both arms | | | |
| | Session 2 | (2)×Simultaneous knee flexion and extension of the 2 legs + Flexion and extension of elbows with adducted shoulders, simultaneous of both arms, (2)×Simultaneous knee flexion and extension of the 2 legs + Flexion and extension of elbows with abducted shoulders, simultaneous of both arms; (2)×Simultaneous hip adduction and abduction of the 2 legs + Flexion and extension horizontal of shoulders, simultaneous of both arms | | | |
| | Session 3 | (2)×Simultaneous hip adduction and abduction of the 2 legs + abbduction and adduction transversal of shoulders, simultaneous of both arms; (2)×Simultaneous knee flexion and extension of the 2 legs + Flexion and extension of el-bows with adducted shoulders, simultaneous of both arms; (2)×Simultaneous knee flexion and extension of the 2 legs + Flexion and extension of elbows with abducted shoulders, simultaneous of both arms | | | |
| Week 4 | Session 1 | (2)×Simultaneous knee flexion and extension of the 2 legs + Flexion and extension of elbows with adducted shoulders, simultaneous of both arms; (2)×Simultaneous hip flexion and extension of the 2 legs + Flexion and extension horizontal of shoulders, simultaneous of both arms; (2)×Simultaneous knee flexion and extension of the 2 legs + Flexion and extension horizontal of shoulders, simultaneous of both arms | 3 min/12–1 4 RPE | 2 min/9 RPE | 30 min |
| | Session 2 | (2)× Simultaneous knee flexion and extension of the 2 legs + Flexion and extension of elbows with adducted shoulders, simultaneous of both arms; (2)×Simultaneous hip flexion and extension of the 2 legs + Flexion and extension horizontal of shoulders, simultaneous of both arms; (2)×Simultaneous hip adduction and abduction of the 2 legs + Flexion and extension horizontal of shoulders, simultaneous of both arms | | | |
| | Session 3 | (2)×Simultaneous knee flexion and extension of the 2 legs + Flexion and extension of elbows with abducted shoulders, simultaneous of both arms; (2)×Simultaneous hip adduction and abduction of the 2 legs + Flexion and extension horizontal of shoulders, simultaneous of both arms; (2)×Simultaneous knee flexion and extension of the 2 legs + Flexion and extension horizontal of shoulders, simultaneous of both arms | | | |

| Weeks | Session | (Sets) x Exercises | Work duration/ Intensity (each exercise) | Rest duration/ Intensity (between sets & exercises) | Total Time (per session) |
|---|---|---|---|---|---|
| Week 5 | Session 1 | (2)×Simultaneous knee flexion and extension of the 2 legs + Flexion and extension of elbows with adducted shoulders, simultaneous of both arms; (2)×Simultaneous hip flexion and extension of the 2 legs + Flexion and extension horizontal of shoulders, simultaneous of both arms; (2)×Simultaneous knee flexion and extension of the 2 legs + Flexion and extension of elbows with abducted shoulders, simultaneous of both arms | | | |
| | Session 2 | (2)×Simultaneous hip adduction and abduction of the 2 legs + abduction and adduction transversal of shoulders, simultaneous of both arms; (2)×Simultaneous hip adduction and abduction of the 2 legs + Flexion and extension horizontal of shoulders, simultaneous of both arms; (2)×Simultaneous knee flexion and extension of the 2 legs + Flexion and extension horizontal of shoulders, simultaneous of both arms | | | |
| | Session 3 | (2)×Simultaneous hip adduction and abduction of the 2 legs + abduction and adduction transversal of shoulders, simultaneous of both arms; (2)×Simultaneous knee flexion and extension of the 2 legs + Flexion and extension of elbows with adducted shoulders, simultaneous of both arms; (2)×Simultaneous hip adduction and abduction of the 2 legs + Flexion and extension horizontal of shoulders, simultaneous of both arms | | | |
| Week 6 | Session 1 | (2)×Simultaneous knee flexion and extension of the 2 legs + Flexion and extension of elbows with abducted shoulders, simultaneous of both arms; (2)×Simultaneous knee flexion and extension of the 2 legs + Flexion and extension horizontal of shoulders, simultaneous of both arms; (2)×Simultaneous hip flexion and extension of the 2 legs + Flexion and extension of elbows with abducted shoulders, simultaneous of both arms | | | |
| | Session 2 | (2)×Simultaneous knee flexion and extension of the 2 legs + Flexion and extension horizontal of shoulders, simultaneous of both arms; (2)×Simultaneous knee flexion and extension of the 2 legs + Abduction and adduction transversal of shoulders, simultaneous of both arms; (2)×Simultaneous hip adduction and abduction of the 2 legs + Flexion and extension of elbows with adducted shoulders, simultaneous of both arms | | | |
| | Session 3 | (2)×Simultaneous knee flexion and extension of the 2 legs + Flexion and extension horizontal of shoulders, simultaneous of both arms; (2)×Simultaneous hip flexion and extension of the 2 legs + Flexion and extension of elbows with abducted shoulders, simultaneous of both arms; (2)×Simultaneous hip flexion and extension of the 2 legs + Flexion and extension of elbows with adducted shoulders, simultaneous of both arms | | | |
| Week 7 | Session 1 | (2)×Simultaneous knee flexion and extension of the 2 legs + Flexion and extension of elbows with abducted shoulders, simultaneous of both arms; (2)×Simultaneous hip flexion and extension of the 2 legs + Flexion and extension of el-bows with abducted shoulders, simultaneous of both arms; (2) × Simultaneous knee flexion and extension of the 2 legs + Abbduction and adduction transversal of shoulders, simultaneous of both arms | | | |
| | Session 2 | (2)×Simultaneous knee flexion and extension of the 2 legs + Flexion and extension horizontal of shoulders, simultaneous of both arms; (2)×Simultaneous hip adduction and abduction of the 2 legs + Flexion and extension of elbows with adducted shoulders, simultaneous of both arms; (2)×Simultaneous hip flexion and extension of the 2 legs + Flexion and extension of elbows with adducted shoulders, simultaneous of both arms | | | |
| | Session 3 | (2)×Simultaneous knee flexion and extension of the 2 legs + Flexion and extension of elbows with abducted shoulders, simultaneous of both arms; (2)×Simultaneous hip flexion and extension of the 2 legs + Flexion and extension of elbows with abducted shoulders, simultaneous of both arms; (2)×Simultaneous hip adduction and abduction of the 2 legs + Flexion and extension of elbows with adducted shoulders, simultaneous of both arms | | | |
| Week 8 | Session 1 | (2)×Simultaneous knee flexion and extension of the 2 legs + Flexion and extension horizontal of shoulders, simultaneous of both arms; (2)×Simultaneous hip flexion and extension of the 2 legs + Flexion and extension of elbows with adducted shoulders, simultaneous of both arms; (2)×Simultaneous knee flexion and extension of the 2 legs + Abbduction and adduction transversal of shoulders, simultaneous of both arms | | | |
| | Session 2 | (2)×Simultaneous knee flexion and extension of the 2 legs + Flexion and extension of elbows with abducted shoulders, simultaneous of both arms; (2)×Simultaneous hip adduction and abduction of the 2 legs + Flexion and extension of elbows with adducted shoulders, simultaneous of both arms; (2)×Simultaneous hip flexion and extension of the 2 legs + Flexion and extension of elbows with adducted shoulders, simultaneous of both arms | | | |
| | Session 3 | (2)×Simultaneous hip flexion and extension of the 2 legs + Flexion and extension of elbows with adducted shoulders, simultaneous of both arms; (2)×Simultaneous knee flexion and extension of the 2 legs + Abbduction and adduction transversal of shoulders, simultaneous of both arms; (2)×Simultaneous hip adduction and abduction of the 2 legs + Flexion and extension of elbows with adducted shoulders, simultaneous of both arms | | | |
**Table 2** (*continued*)

| Weeks | Session | (Sets) × Exercises | Work duration/ Intensity (each exercise) | Rest duration/ Intensity (between sets & exercises) | Total Time (per session) |
|---|---|---|---|---|---|
| Week 9 | Session 1 | (2) × Simultaneous knee flexion and extension of the 2 legs + Abbduction and adduction transversal of shoulders, simultaneous of both arms; (2) × Simultaneous hip adduction and abduction of the 2 legs + Flexion and extension of elbows with adducted shoulders, simultaneous of both arms; (2) × Simultaneous hip flexion and extension of the 2 legs + Flexion and extension of elbows with adducted shoulders, simultaneous of both arms | | | |
| | Session 2 | (2) × Simultaneous knee flexion and extension of the 2 legs + Flexion and extension horizontal of shoulders, simultaneous of both arms; (2) × Simultaneous hip flexion and extension of the 2 legs + Flexion and extension of elbows with adducted shoulders, simultaneous of the 2 legs + Flexion and extension of elbows with adducted shoulders, simultaneous of both arms | | | |
| | Session 3 | (2) × Simultaneous knee flexion and extension of the 2 legs + Flexion and extension of elbows with abducted shoulders, simultaneous of both arms; (2) × Simultaneous knee flexion and extension of the 2 legs + Flexion and extension horizontal of shoulders, simultaneous of both arms; (2) × Simultaneous knee flexion and extension of the 2 legs + Abbduction and adduction transversal of shoulders, simultaneous of both arms | 3 min/15-16 RPE | 2 min/9 RPE | 30 min |
| Week 10 | Session 1 | (2) × Simultaneous knee flexion and extension of the 2 legs + Flexion and extension of elbows with abducted shoulders, simultaneous of both arms; (2) × Simultaneous hip flexion and extension of the 2 legs + Flexion and extension of elbows with abducted shoulders, simultaneous of both arms; (2) × Simultaneous hip flexion and extension of the 2 legs + Flexion and extension of elbows with adducted shoulders, simultaneous of both arms | | | |
| | Session 2 | (2) × Simultaneous knee flexion and extension of the 2 legs + Flexion and extension horizontal of shoulders, simultaneous of both arms; (2) × Simultaneous knee flexion and extension of the 2 legs + Abbduction and adduction transversal of shoulders, simultaneous of both arms; (2) × Simultaneous hip flexion and extension of the 2 legs + Flexion and extension of elbows with adducted shoulders, simultaneous of both arms | | | |
| | Session 3 | (2) × Simultaneous knee flexion and extension of the 2 legs + Flexion and extension of elbows with abducted shoulders, simultaneous of both arms; (2) × Simultaneous knee flexion and extension of the 2 legs + Abbduction and adduction transversal of shoulders, simultaneous of both arms; (2) × Simultaneous hip adduction and abduction of the 2 legs + Flexion and extension of elbows with adducted shoulders, simultaneous of both arms | | | |

**Notes.**

RPE, rating of perceived exertion.

**Table 3 Changes in anthropometric and biochemical variables in both groups.**

| Variables | Experimental group ($n = 16$) | | | | Control group ($n = 11$) | | | |
| --- | --- | --- | --- | --- | --- | --- | --- | --- |
| | Pre-test | Post-test | Δ | ES | Pre-test | Post-test | Δ | ES |
| Weight (kg) | 81.2 ± 8.2 | 77.3 ± 7.6[†] | −4.9 ± 1.6 | 0.493 | 78.3 ± 8.1 | 79.4 ± 8.4[†] | 1.4 ± 0.9 | 0.133 |
| BMI (kg m$^{-2}$) | 29.9 ± 3.1 | 28.9 ± 3.0[†] | −3.4 ± 1.5 | 0.328 | 30.1 ± 3.1 | 30.5 ± 3.2[†] | 1.4 ± 0.9 | 0.127 |
| BF (%) | 37.4 ± 4.0 | 35.3 ± 3.4[†] | −5.5 ± 3.0 | 0.566 | 36.4 ± 3.3 | 36.6 ± 3.9 | 0.3 ± 3.7 | 0.055 |
| Glycemia (mmol L$^{-1}$) | 5.19 ± 0.38 | 4.88 ± 0.16[†,‡] | −5.7 ± 5.5 | 1.063 | 5.37 ± 0.24 | 5.42 ± 0.27[†] | 0.9 ± 1.3 | 0.196 |
| TC (mmol L$^{-1}$) | 4.65 ± 0.65 | 4.43 ± 0.55 | −4.6 ± 3.3 | 0.365 | 4.29 ± 0.50 | 4.49 ± 0.34 | 5.4 ± 7.9 | 0.468 |
| HDL-C (mmol L$^{-1}$) | 3.06 ± 0.39 | 3.31 ± 0.35[†] | 8.9 ± 8.4 | 0.675 | 3.13 ± 0.23 | 3.04 ± 0.23 | −2.8 ± 5.0 | 0.391 |
| LDL-C (mmol L$^{-1}$) | 1.18 ± 0.57[a] | 0.83 ± 0.38 | −27.2 ± 14.0 | 0.723 | 0.76 ± 0.38 | 1.10 ± 0.30 | 112.2 ± 184.7 | 0.993 |
| TG (mmol L$^{-1}$) | 0.97 ± 0.27 | 0.68 ± 0.23[†] | −26.4 ± 26.2 | 1.156 | 0.88 ± 0.38 | 0.79 ± 0.20 | −0.4 ± 28.7 | 0.296 |

Notes.

Values are given as means ± standard deviation.

BMI, body mass index; BF, body fat; TC, total cholesterol; HDL-C, high-density lipoprotein cholesterol; LDL-C, low-density lipoprotein cholesterol; TG, triglyceride; ES, effect size; Δ, pre- to-post change percentage.

[†] A significant difference when comparing pre-test and Post-test.

[‡] Significantly different from CG at Post-test.

[a] Significantly different from CG at pre-test.

The statistical significance level was set at $p \leq 0.05$.

**Table 4 Changes in cardiovascular variables in both groups.**

| Variables | Experimental group ($n = 16$) | | | | Control group ($n = 11$) | | | |
| --- | --- | --- | --- | --- | --- | --- | --- | --- |
| | Pre-test | Post-test | Δ | ES | Pre-test | Post-test | Δ | ES |
| RHR (bpm) | 76.9 ± 6.5 | 74.8 ± 6.0[†] | −2.7 ± 1.2 | 0.336 | 77.7 ± 3.3 | 78.5 ± 3.1[†] | 0.8 ± 1.0 | 0.250 |
| RSBP (mm HG) | 117.4 ± 9.1 | 117.7 ± 4.6 | 0.5 ± 3.9 | 0.042 | 121.5 ± 3.6 | 121.6 ± 8.3 | 0.0 ± 1.2 | 0.016 |
| RDBP (mm HG) | 72.4 ± 9.9 | 73.4 ± 7.5 | 1.9 ± 4.5 | 0.114 | 73.3 ± 5.6 | 72.5 ± 5.4 | −1.1 ± 1.6 | 0.145 |

Notes.

Values are given as means ± standard deviation.

RHR, resting heart rate; RSBP, resting systolic blood pressure; RDBP, resting diastolic blood pressure; ES, effect size; Δ, pre- to-post change percentage.

[†] A significant difference when comparing pre-test and Post-test.

The statistical significance level was set at $p \leq 0.05$.

**Table 5 Changes in explosive strength variables in both groups.**

| Variables | Experimental group ($n = 16$) | | | | Control group ($n = 11$) | | | |
| --- | --- | --- | --- | --- | --- | --- | --- | --- |
| | Pre-test | Post-test | Δ | ES | Pre-test | Post-test | Δ | ES |
| CMJ (cm) | 19.2 ± 4.4c | 20.0 ± 4.5[†,‡] | 4.4 ± 1.7 | 0.180 | 14.5 ± 3.9 | 14.4 ± 3.8 | −0.3 ± 1.3 | 0.026 |
| SJ (cm) | 13.2 ± 4.8 | 14.2 ± 4.9[†] | 8.0 ± 5.0 | 0.206 | 14.1 ± 3.6 | 14.3 ± 3.5 | 1.0 ± 1.0 | 0.056 |
| MBT (m) | 2.2 ± 0.4 | 3.0 ± 0.6[†,‡] | 35.1 ± 15.6 | 1.512 | 2.4 ± 0.2 | 2.4 ± 0.2 | −0.2 ± 6.3 | 0.043 |

Notes.

Values are given as means ± standard deviation.

CMJ, countermovement jump; SJ, squat jump; MBT, seated medicine ball throw test; ES, effect size; Δ, pre- to-post change percentage.

[†] A significant difference when comparing pre-test and post-test.

[‡] Significantly different from CG at post-test.

[a] Significantly different from CG at pre-test.

The statistical significance level was set at $p \leq 0.05$.

# RESULTS

All but MBT scores were normally distributed. Except for age ($p = 0.011$; ES = 0.972, large), height ($p = 0.043$; ES = 0.714, medium), LDL-C ($p = 0.043$; ES = 0.963, large) and CMJ ($p = 0.009$; ES = 1.117, large), CG showed no significant differences in terms of anthropometric, biochemical, cardiovascular, and explosive strength (*i.e.*, SJ and MBT) measurements compared to EG at baseline (all $p > 0.05$) (Tables 1, 3, 4 and 5).

There was a main effect for time in body weight ($F = 41.281$; $p < 0.0001$; ES = 1.221, large), BMI ($F = 15.621$; $p = 0.001$; ES = 0.736, medium), %BF ($F = 13.848$; $p = 0.001$; ES = 0.690, medium), fasting glycemia ($F = 7.276$; $p = 0.001$; ES = 0.482, small), HDL-C

($F = 5.344$; $p = 0.029$; ES $= 0.401$, small), TG ($F = 10.161$; $p = 0.004$; ES $= 0.582$, medium), RHR ($F = 16.198$; $p < 0.0001$; ES $= 0.750$, medium) and SJ ($F = 57.847$; $p < 0.0001$; ES $= 1.451$, large). EG improved its body weight, BMI, %BF, fasting glycemia, LDL-C, HDL-C, TG, RHR and explosive strength measurements over the 10-week intervention (all $p < 0.05$; ES $= 0.180$–$1.512$, trivial to large) (Tables 3, 4 and 5). In contrast, the body weight, BMI, fasting glycemia, LDL-C and RHR increased in CG after the intervention period (all $p < 0.05$; ES $= 0.127$–$0.993$, trivial to large). %BF, fasting glycemia, TG, CMJ, SJ and MBT measurements remained unchanged in CG over the intervention (Tables 3, 4 and 5). Also, no main effect for time was identified in TC ($F = 0.050$; $p = 0.826$; ES $= 0$, trivial), RSBP ($F = 0.073$; $p = 0.790$; ES $= 0$, trivial) and RDBP ($F = 0.054$; $p = 0.817$; ES $= 0$, trivial) measures (Table 4).

For fasting glycemia, there was a main effect for condition ($F = 13.732$; $p = 0.252$; ES $= 0.687$, medium), and a significant interaction effect was observed between time and condition ($F = 13.893$; $p = 0.001$; ES $= 0.691$, medium), with EG showing lower post-test values than CG ($p < 0.0001$; ES $= 2.559$, large) (Table 3).

For LDL-C and CMJ, there was a main effect for condition ($F = 72.901$ and $35.087$; both $p < 0.0001$; ES $= 1.663$ and $1.145$, large, respectively), with EG eliciting better scores than CG at post-test (both $p < 0.0001$; ES $= 0.971$ and $1.323$, large) (Table 5). Moreover, MBT performance was significantly higher in EG compared to CG at post-test ($p = 0.003$; ES $= 1.311$, large) (Table 5).

The results of the analysis on the body weight ($F = 128.630$; $p < 0.0001$; ES $= 2.174$, large), BMI ($F = 89.867$; $p < 0.0001$; ES $= 1.814$, large), %BF ($F = 18.246$; $p < 0.0001$; ES $= 0.779$, medium), TC ($F = 24.668$; $p < 0.0001$; ES $= 0.936$, large) RHR ($F = 55.734$; $p < 0.0001$; ES $= 1.424$, large), and SJ ($F = 33.156$; $p < 0.0001$; ES $= 1.091$, large) showed a significant interaction between time and condition. In contrast, no main effect for condition was observed (body weight: $F = 0.019$; $p = 0.892$; ES $= 0$, trivial; BMI: $F = 0.577$; $p = 0.454$; ES $= 0$, trivial; %BF: $F = 0.008$; $p = 0.930$; ES $= 0$, trivial; TC: $F = 24.668$; $p < 0.0001$; ES $= 0.936$, large; RHR: $F = 1.374$; $p = 0.252$; ES $= 0.118$, trivial; SJ: $F = 0.089$; $p = 0.768$; ES $= 0$, trivial) (Tables 3, 4 and 5).

No statistical interactions nor main effects for condition were observed in HDL-C ($F = 3.096$ and $3.516$; $p = 0.090$ and $0.072$; ES $= 0.279$ and $0.305$, small, respectively), TG ($F = 2.637$ and $0.015$; $p = 0.117$ and $0.902$; ES $= 0.246$ and $0$, trivial to small, respectively), RSBP ($F = 0.022$ and $2.074$; $p = 0.883$ and $0.883$; ES $= 0$ and $0.199$, trivial, respectively), and RDBP ($F = 3.228$ and $0.000$; $p = 0.084$ and $0.989$; ES $= 0.287$ and $0$, trivial to small, respectively) measurements (Tables 3 and 4).

## DISCUSSION

The present study assessed the effects of a 10 week water-based aerobic training on anthropometric, biochemical, cardiovascular and explosive strength parameters in young obese and overweight women. The results showed that 10 weeks of water-based aerobic training led to an improved anthropometric and biochemical parameters with changing in lipid profile values, explosive strength and resting heart rate measures, which partially

supporting our hypothesis. There was also an effect of condition for LDL-C, fasting glycemia and explosive strength (*i.e.,* CMJ, MBT) performances with significant improvements in EG compared to CG. In contrast, the body weight, BMI, LDL-C, RHR, and fasting glycemia increased after the intervention in CG.

Our results have shown a significant decrease of body weight, BMI and %BF in EG. These findings are in line with previous studies showing the beneficial effects of water-based physical exercises on anthropometric parameters in overweight and obese women (*Bielec, Kwasna & Gaworska, 2017*; *Penaforte et al., 2015*). *Bielec, Kwasna & Gaworska (2017)* have reported that six months of aqua power aerobics changed body composition in middle-aged overweight women. Similarly, *Penaforte et al. (2015)* demonstrated that two months of continuous water exercise improved body weight, BMI, %BF, and arm and hip circumferences. The physical properties of water decrease the physical conditioning of obese individuals who have limitations of mobility and encourage them to engage in regular physical activity (*Torres-Ronda & Del Alcázar, 2014*). Likewise, regular water exercise induces an increase in their energy expenditure and consequently an improvement in their anthropometrics (*Bielec, Kwasna & Gaworska, 2017*).

In the present study, the results showed a slight increase in HDL-C levels (8.9%), and a decrease in LDL-C (−27.2%) and TG levels (26.4%) after 10 weeks of water exercise intervention but no significant changes in TC levels (−4.6%). In addition, a significant decrease in fasting glycemia levels was recorded. In accordance with our results, *Kasprzak & Pilaczyńska-Szcześniak (2014)* found a significant decrease in LDL-C and TC and no changes were observed in HDL-C levels in women with abdominal obesity after a 12-week aqua aerobics program. *Penaforte et al. (2015)* observed no changes in lipid profile variables among middle-aged obese women after 8 weeks of water aerobics program. These conflicting results suggest that the response of lipid profile and glycemia variables to a water training intervention is influenced by the duration of the program, the type of exercise, its intensity and frequency, and the health status of participants. In fact, short-term interventions (*e.g.,* less than 8 weeks) may not provide enough time for the body to undergo significant metabolic adaptations, whereas longer programs allow for cumulative improvements in insulin sensitivity and lipid metabolism (*Patel et al., 2017*; *Kraus et al., 2002*). Moreover, the specific exercises performed during water-based training, whether aerobic or resistance-focused, can differently impact metabolic health (*Torres-Ronda & Del Alcázar, 2014*). Aerobic exercises are particularly effective in enhancing cardiovascular fitness, improving insulin sensitivity, and reducing LDL-C and TG (*Torres-Ronda & Del Alcázar, 2014*; *Alberton et al., 2013b*). On the other hand, resistance exercises might contribute to increased muscle mass, which can improve basal metabolic rate and glycemic control (*Torres-Ronda & Del Alcázar, 2014*). The intensity and frequency of the exercise are crucial determinants of the metabolic benefits derived from the intervention (*Pescatello et al., 2014*). Higher intensity exercises performed more frequently tend to produce more pronounced improvements in lipid profiles and glycemic control. Moderate to high-intensity aerobic exercises are particularly effective in enhancing insulin sensitivity, reducing TG, and increasing HDL-C levels (*Pescatello et al., 2014*). ACSM recommends moderate to high-intensity exercise performed three to five times per week for optimal

improvements in lipid and glycemic profiles (*Pescatello et al., 2014*). It is important to note that the interval training prescribed in our study showed gains at increasing intensities from moderate to high. This could have led to the positive responses shown in the majority of biochemical outcomes over time by continuously inducing new training adaptations. The initial health status of participants, including factors such as baseline metabolic health, insulin sensitivity, and lipid levels, can influence how their bodies respond to the exercise intervention. Participants with poorer baseline health (*e.g.*, higher baseline TG or insulin resistance) may experience more significant improvements due to the greater potential for metabolic adaptations compared to those with better baseline metabolic health (*Lakka & Laaksonen, 2007*). The improved lipid profile variables observed in our participants could be explained by the increase of the activity of enzymes responsible for the breaking down of TG in muscles and adipocytes leading to a reduction of blood TG levels (*Welsh et al., 2024*; *Franczyk et al., 2023*; *Church & Martin, 2018*; *Pescatello et al., 2014*). In addition, physical activity had significant effects on glucose levels in individuals with overweight or obesity (*Welsh et al., 2024*). This may be associated with weight loss and a reduction in %BF or an increase in lipid oxidation capacity in these obese individuals (*Franczyk et al., 2023*; *De Sousa Fernandes et al., 2022*).

Regarding RHR, our EG recorded a decrease while CG showed an increase after the intervention period. However, no effect of time was for RSBP and RDBP in both groups. Previous studies about the effectiveness of aquatic exercises on cardiovascular fitness revealed that training in water media could significantly enhance RHR in obese and overweight women at different age stages (*Abadi et al., 2017*; *Bielec, Kwasna & Gaworska, 2017*; *Bielec, Kwasna & Gaworska, 2017*; *Piotrowska-Calka, 2010*). These positive results might be explained by the exercise itself and because of the high temperature usually found in the swimming pools (*Pereira Neiva et al., 2018*). Perhaps there was an adaptation of the sympathetic and parasympathetic nervous system, with the reduction of the first and stimulation of the second, a change that has already been shown when the bodies are immersed into higher temperatures (*Nahimura et al., 2008*). The lack of change in resting blood pressure variables in our EG was not expected. It is well known that regular land or water exercise enhances blood vessel elasticity and function which contributes to improved blood flow and potentially lower blood pressure (*Green & Smith, 2018*). In addition, the pressure in water could enhance blood circulation and reduce blood pressure (*Igarashi & Nogami, 2018*). The reasons that can explain the unchanged resting blood pressure measures after the intervention, particularly RSBP, might be related to the age and cardiovascular status of our subjects who were young and did not suffer from arterial hypertension or other cardiovascular diseases.

Furthermore, our findings demonstrated that a 10-week water-based aerobic training significantly improved explosive strength in young obese and overweight women, whereas the CG failed to yield significant improvements in any of the variables. Similarly, earlier reports about the effectiveness of aquatic exercises on explosive strength revealed that training in water media could significantly increase upper and lower body strength in different populations (*Nosrani et al., 2023*; *Scheer et al., 2020*; *Pereira Neiva et al., 2018*). *Nosrani et al. (2023)* studied a group of overweight healthy older adults of both sexes

who aquatically trained over twenty-eight weeks and found similar results to those in this study for the explosive strength of upper and lower limbs. *Pereira Neiva et al. (2018)* have reported that 12 weeks of a water aerobics program in healthy overweight adults of both sexes enhanced the explosive strength of the upper limbs but not of the lower limbs. It was also revealed that aquatic exercise training with integrated aerobic and resistance exercises improved strength and aerobic fitness in people with diabetes type 2 (*Scheer et al., 2020*). Moreover, *Scheer et al. (2021)* showed a significant improvement in aerobic capacity and leg strength in people with stable coronary heart disease. Water training has an important impact on both upper and lower body strength due to the resistance and buoyancy properties of water (*Torres-Ronda & Del Alcázar, 2014*). Water's buoyancy reduces the impact on joints while providing resistance during movements for lower body strength (*Torres-Ronda & Del Alcázar, 2014*). Water resistance requires the muscles in the arms, shoulders, and chest to work against the resistance in all directions, promoting not only strength but also endurance (*Torres-Ronda & Del Alcázar, 2014*). Water serves as resistance for motions typically performed on land (*e.g.*, walking, jogging, or jumping). In our study, the aquatic program was designed so that exercises replicate daily activities. Water exercise training offers benefits comparable to other exercise forms, reducing the risk of muscle and joint injuries, and improving cardiovascular and muscle fitness, anthropometrics and blood lipids in obese persons (*Abadi, 2023*).

The increased body weight, BMI, LDL-C, RHR and fasting glycemia level in CG can be explained through several physiological and lifestyle-related factors (*e.g.*, energy imbalance, lack of regular physical activity) (*Welsh et al., 2024*). These changes are typically associated with a sedentary lifestyle and the lack of regular physical activity, which CG presumably maintained. The lack of physical activity can lead to an energy imbalance where caloric intake exceeds caloric expenditure, resulting in weight gain and an increase in BMI (*Welsh et al., 2024*; *Church & Martin, 2018*). Physical activity (*e.g.*, water-based aerobic training) might help in managing weight by increasing energy expenditure (*Welsh et al., 2024*; *Church & Martin, 2018*). The increase in body weight and BMI among CG can be attributed to the continued absence of such energy expenditure (*Welsh et al., 2024*; *Church & Martin, 2018*). A higher RHR in CG could indicate a lower cardiovascular fitness level, as physical inactivity is known to reduce cardiovascular efficiency (*Welsh et al., 2024*). Regular aerobic exercise, like swimming or water aerobics, improves heart health and can lower RHR by enhancing the heart's efficiency (*Cornelissen & Smart, 2013*). Furthermore, an increase in fasting blood glucose and LDL-C levels in CG might reflect a lack of beneficial effects from physical activity on insulin sensitivity and glucose metabolism as well as blood lipid metabolism (*Welsh et al., 2024*; *Franczyk et al., 2023*). Aerobic exercise is known to improve insulin sensitivity and help regulate blood glucose levels by enhancing the muscle's ability to use glucose (*Welsh et al., 2024*; *Colberg et al., 2010*). Furthermore, such activity accelerates the transfer, breakdown and excretion of lipids and reduces fasting or postprandial TG (*Franczyk et al., 2023*). Besides the aforementioned changes, it reduces TC and LDL-C levels (*Franczyk et al., 2023*).

This investigation faces limitations that should be considered when interpreting the results. Firstly, the lack of nutritional monitoring, crucial for assessing and

enhancing dietary habits for the healthy lifestyle in obese persons. Secondly, the absence of biomechanical, electrophysiological, inflammatory markers and muscle damage assessments precluded a deeper understanding of the exercise intervention's effects and underlying mechanisms (*e.g.*, muscle architecture and muscle action) in preventing metabolic and cardiovascular diseases. The third limitation involves sex and age, making it speculative to generalize our findings to other populations (*e.g.*, middle-aged obese women or men, overweight healthy older adults with cardiovascular diseases). This is undoubtedly a limiting factor and may introduce bias. While our study demonstrates the efficacy of water-based aerobic training in improving certain health and fitness parameters in young obese and overweight women, it also underscores the need for further research. Investigating the integration of aquatic exercise with dietary adjustments, expanding the study population, and exploring varied exercise regimens are critical steps toward developing more holistic and inclusive health management strategies. Finally, there was a notable disparity in age and height between groups with EG being older and taller than CG. This discrepancy could potentially act as a confounding factor. However, the lack of a significant difference in the pre-test anthropometric, biochemical, cardiovascular, and explosives strength values between groups can help mitigate the impact of these confounders.

## CONCLUSIONS

The study found that a 10-week water-based aerobic training program led to significant improvements in body composition, cardiovascular health, and explosive strength in young overweight and obese women. These findings suggest that water-based aerobic exercise is effective in improving key health markers and should be included in obesity management programs. Our findings emphasize the long-term benefits of such exercise in sustaining weight loss and reducing obesity-related health risks. Additionally, the improvements in explosive strength indicate that this form of exercises can enhance both cardiovascular and muscular fitness, making it a valuable component of fitness programs aimed at improving overall health and quality of life for overweight and obese individuals. This study supports the inclusion of a structured, moderate-to-high intensity water-based aerobic exercise in public health strategies, particularly for this population, as it offers a safe, effective, and enjoyable way to promote physical activity and improve health outcomes.

### Funding

The authors received no funding for this work.

### Competing Interests

Prof. Yung-Sheng Chen is an Academic Editor for PeerJ.

### Author Contributions

- Imen Ben Cheikh conceived and designed the experiments, performed the experiments, prepared figures and/or tables, and approved the final draft.
- Hamza Marzouki conceived and designed the experiments, authored or reviewed drafts of the article, and approved the final draft.
- Okba Selmi conceived and designed the experiments, authored or reviewed drafts of the article, and approved the final draft.
- Bilel Cherni performed the experiments, prepared figures and/or tables, and approved the final draft.
- Siwar Bouray performed the experiments, prepared figures and/or tables, and approved the final draft.
- Ezdine Bouhlel performed the experiments, prepared figures and/or tables, and approved the final draft.
- Anissa Bouassida analyzed the data, authored or reviewed drafts of the article, and approved the final draft.
- Beat Knechtle analyzed the data, authored or reviewed drafts of the article, and approved the final draft.
- Yung-Sheng Chen analyzed the data, authored or reviewed drafts of the article, and approved the final draft.

### Human Ethics

The following information was supplied relating to ethical approvals (i.e., approving body and any reference numbers):

Research Ethics Committee of the University of Jendouba granted Ethical approval to carry out the study within its facilities (Ethical Application Ref:014/2018).

### Clinical Trial Ethics

The following information was supplied relating to ethical approvals (i.e., approving body and any reference numbers):

The University of Jendouba granted Ethical approval to conduct the study within its facilitites.

### Data Availability

The raw measurements are available in the Supplementary File.

## Clinical Trial Registration

The following information was supplied regarding Clinical Trial registration:
    NCT06371105

## Supplemental Information

Supplemental information for this article can be found online at http://dx.doi.org/10.7717/peerj.19020#supplemental-information.

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
