# Peer review of "Effect of water-based aerobic training on anthropometric, biochemical, cardiovascular, and explosive strength parameters in young overweight and obese women: a randomized controlled trial"

_PeerJ, doi:10.7717/peerj.19020_

## Round 0.1 · original submission · Major Revisions

· Academic Editor

Major Revisions

Dear Authors:

Thank you for submitting your manuscript to PeerJ Journals. After review, we consider Major Reviews. Please attend to the reviewers´ comments.

I hope to read your reviewed manuscript forward.

All the best

Dr. Manuel Jimenez

Reviewer 1 ·

Basic reporting

Thank you for the opportunity to review an interesting article.
1. Please sort out the information because in the abtract the authors write that the aim of the study is to see if training has an effect and in the introduction the authors write that the research hypothesis: ‘We hypothesised that 10 weeks of water-based aerobic training would yield improvements in health status and explosive strength’.
A research hypothesis and an objective are not the same thing. The research hypothesis shall be tested statistically and the objective shall not.

2. Materials & Methods; Participants - the authors begin with information on statistical analysis.
Please describe: the group of study participants, what was the inclusion and exclusion criterion, what parameters were studied and how, how the study was randomised and finally describe the statistical analysis i.e. what statistical package was used with add reference to the literature, what tests were carried out and for what purpose?
Isolate in separate paragraphs so that it is visible and legible.

3. Line 137-143 - include this information in the statistical analysis.

4. Tables - in each table add a description of the value for significance - p-value, the last columns in the tables for ANOVA are unreadable, I propose leaving only the values for F and p-value.

5. Discussion – suggests enriching the discussion with references to as recent publications as possible, e.g. DOI: 10.3389/fpubh.2024.1371420


6. Limitations - it is worth adding a description of the limitations of the study after the discussion, as this study had several limitations.

Experimental design

Thank you for the opportunity to review an interesting article.
1. Please sort out the information because in the abtract the authors write that the aim of the study is to see if training has an effect and in the introduction the authors write that the research hypothesis: ‘We hypothesised that 10 weeks of water-based aerobic training would yield improvements in health status and explosive strength’.
A research hypothesis and an objective are not the same thing. The research hypothesis shall be tested statistically and the objective shall not.

2. Materials & Methods; Participants - the authors begin with information on statistical analysis.
Please describe: the group of study participants, what was the inclusion and exclusion criterion, what parameters were studied and how, how the study was randomised and finally describe the statistical analysis i.e. what statistical package was used with add reference to the literature, what tests were carried out and for what purpose?
Isolate in separate paragraphs so that it is visible and legible.

3. Line 137-143 - include this information in the statistical analysis.

4. Tables - in each table add a description of the value for significance - p-value, the last columns in the tables for ANOVA are unreadable, I propose leaving only the values for F and p-value.

5. Discussion – suggests enriching the discussion with references to as recent publications as possible, e.g. DOI: 10.3389/fpubh.2024.1371420


6. Limitations - it is worth adding a description of the limitations of the study after the discussion, as this study had several limitations.

Validity of the findings

Thank you for the opportunity to review an interesting article.
1. Please sort out the information because in the abtract the authors write that the aim of the study is to see if training has an effect and in the introduction the authors write that the research hypothesis: ‘We hypothesised that 10 weeks of water-based aerobic training would yield improvements in health status and explosive strength’.
A research hypothesis and an objective are not the same thing. The research hypothesis shall be tested statistically and the objective shall not.

2. Materials & Methods; Participants - the authors begin with information on statistical analysis.
Please describe: the group of study participants, what was the inclusion and exclusion criterion, what parameters were studied and how, how the study was randomised and finally describe the statistical analysis i.e. what statistical package was used with add reference to the literature, what tests were carried out and for what purpose?
Isolate in separate paragraphs so that it is visible and legible.

3. Line 137-143 - include this information in the statistical analysis.

4. Tables - in each table add a description of the value for significance - p-value, the last columns in the tables for ANOVA are unreadable, I propose leaving only the values for F and p-value.

5. Discussion – suggests enriching the discussion with references to as recent publications as possible, e.g. DOI: 10.3389/fpubh.2024.1371420


6. Limitations - it is worth adding a description of the limitations of the study after the discussion, as this study had several limitations.

Additional comments

Thank you for the opportunity to review an interesting article.
1. Please sort out the information because in the abtract the authors write that the aim of the study is to see if training has an effect and in the introduction the authors write that the research hypothesis: ‘We hypothesised that 10 weeks of water-based aerobic training would yield improvements in health status and explosive strength’.
A research hypothesis and an objective are not the same thing. The research hypothesis shall be tested statistically and the objective shall not.

2. Materials & Methods; Participants - the authors begin with information on statistical analysis.
Please describe: the group of study participants, what was the inclusion and exclusion criterion, what parameters were studied and how, how the study was randomised and finally describe the statistical analysis i.e. what statistical package was used with add reference to the literature, what tests were carried out and for what purpose?
Isolate in separate paragraphs so that it is visible and legible.

3. Line 137-143 - include this information in the statistical analysis.

4. Tables - in each table add a description of the value for significance - p-value, the last columns in the tables for ANOVA are unreadable, I propose leaving only the values for F and p-value.

5. Discussion – suggests enriching the discussion with references to as recent publications as possible, e.g. DOI: 10.3389/fpubh.2024.1371420


6. Limitations - it is worth adding a description of the limitations of the study after the discussion, as this study had several limitations.

Annotated reviews are not available for download in order to protect the identity of reviewers who chose to remain anonymous.

·

Basic reporting

The introduction part needs to be revised.

Experimental design

Some issues needs to be revised

Validity of the findings

The rationale and benefit to literature is not clearly stated

Additional comments

Comments

Abstract

1.Line 46, Please use 'it is' and check the rewriting throughout your entire paper.

2.Lines 46-48, The background should be revised to clearly state the problem and highlight the research gap.

3.Line 52, “randomized” assigned into experimental, and provide participants' BMI information

4.Line 57, “resting heart rate-RHR” to “resting heart rate (RHR)”

5.Line 55, List anthropometrics components

6.Lines 58-61, The results section should be revised to ensure consistency with the conclusions. Consider incorporating effect sizes to better reflect changes and add between-group comparisons.

7.Line 59, line 63, “athletic performances”, “body composition” inconsistent with your title

Introduction

The introduction part needs to be revised. Here are some ideas: 1) introduce the seriousness of the problem (obesity and overweight); 2) Highlight the advantages of water-based training compared to land-based training; 3) Review the literature on water-based training, its application among overweight populations or other participants, why this type of training was chosen for young overweight and obese women, and why factors such as anthropometric, biochemical, cardiovascular, and explosive strength parameters are important for them."

Materials & Methods

1.Line147, How do you define young age as 20 and 35 years old?

2.Add study location

3.Line 148, How do you define body mass index (BMI) indicating overweight and obese status as between 25.0 and 34.9 kg/m²

4.Line 149, Participants who have prior experience with water-based exercise routines, please provide more detail.

5.Line 182, why 12 weeks here, may confuse reader

6.Lines 193-194,I suggest changing 'physical fitness (upper and lower limbs explosive strength)' to 'explosive strength (CMJ...)

7.Line 237, "Power" and "strength" are indeed different terms in the context of physical fitness, I suggest defining explosive strength in your study

8.Line 260. Abbreviate SJ and CMJ after the first use.

9.line 263, Describe the content of the control group

10.Line 265, The water-based aerobic program was adopted from the protocol previously used by Costa et al. (2018, 2020), but these protocols were focused on dyslipidemic women (aged 40–50 years). How can we ensure it is suitable for young obese and overweight women?

11.How do you consider the participants' monthly menstrual cycles in the intervention?

Results

1.Line 306, Please report the dropout rate and whether all subjects for whom outcome measures received the training intervention, as well as any adverse effects

2.Line 314-316, rewrite this sentence

3.Line 310, Use the abbreviation 'EG'. Please check the entire paper for abbreviations, such as on line 326, 347, 370, and 417. Check carefully throughout.

4.Line 324-329, How about interactions between the time and group results?

5.Overall, I'd like to suggest that you rewrite the results section; it is not easy to read.

Discussion

1.Line 338, Change “relative to CG” to “compare to CG”, and check the entire paper.

2.Line 342, Explain why the CG increased slightly but significantly in their body mass and BMI.

3.Lines 361-364, Explain how these factors influence the water training intervention.

4.Line 366, check punctuation

5.Line 402, This section discusses strength; this statement seems unnecessary.

6.Lines 418-420, line 423, add reference

7.Line 446, I suggest removing 'thrice a week' and revising 'short-term' to 10 weeks.

8.Line 449-453, Future research may explore this aquatic exercise approach across different intensities and durations to improve factors that did not change significantly.

9.The conclusion section should be revised to directly address the research question. Additionally, it should include practical implications that outline how the findings can be applied in real-world settings.

10.In Table 1, I suggest dividing it into two tables: one for the baseline characteristics of the water training and control groups, and one for the baseline and post-intervention data. Alternatively, you could use figures.

11.Why did you not compare the age and height of the two groups at baseline? If the participants in the two groups differ, how can you be sure that age and height do not affect the results? As this is a randomized controlled trial, have you controlled for other confounding factors

12.In Table 2, the training protocol is not clearly described. Please detail the volume (e.g., sets, repetitions, rest periods). Did all exercises remain the same from weeks 1-5 or 6-10? Why was the validity of the training protocol not verified?

13.Please rearrange all the tables to make them clearer for the reader.

14.In the paper, please ensure consistency in all key terms used, such as 'athletic measures,' 'explosive strength,' 'power,' and 'physical fitness'; 'body composition,' and 'anthropometric measurements.'

---

## Round 0.2 · Major Revisions

· Academic Editor

Major Revisions

Dear Authors:

Thank you for submitting your manuscript titled: "Effect of water-based aerobic training on anthropometric, biochemical, cardiovascular, and explosive strength parameters in young overweight and obese women: A randomized controlled trial"; after a well-manned review, we consider Major Reviews.

Please, attend reviewer´s comments.

Regrads

Dr. Manuel Jiménez

Reviewer 1 ·

Basic reporting

Thank you for the opportunity to review an interesting article. The authors have corrected the manuscript.

Experimental design

Thank you for the opportunity to review an interesting article. The authors have corrected the manuscript.

Validity of the findings

Thank you for the opportunity to review an interesting article. The authors have corrected the manuscript.

·

Basic reporting

Comments

Line 64, explosive strength (0.035≤ p< 0.000), please revise to (0.035≤ p< 0.001)

Lines 98-99,lines 102-103, Please use the latest references

Line 179, remove however, or change to another conjunction

Line 164, line 175, correct body mass index to BMI, check the whole paper

Line 155-156, This statement seems unnecessary in the introduction

Line 179-180, EG, CG, use the full name in the text when used for the first time.

Include your consideration of the participants' monthly menstrual cycles in the intervention section of your paper, rather than just answering my question

Line 215, change “counter movement jump” to “countermovement jump”

Line 641, r is what?

Line 354, how about SJ

Line 517, check “exersice”

Lines 507-509, It is unnecessary to restate the results

line 511, check “parameteres”

Line 383, check “an”

Lines 418-421, add references

Line 466-470, I suggest moving this sentence to the limitations section

Line 511, check emphasizes

Table 2, Place the detail exercises in the table instead of listing them as footnotes

I’d like to suggest rearranging all tables: Table 1 (Participant characteristics at baseline) and Tables 3-5 (Changes in anthropometric, biochemical, cardiovascular, and explosive strength parameters).

Table 5, Change “Athletic measures” to explosive strength

Figure 1, Please use the original CONSORT table and add a reference

In tables, why were some marks placed on the pretest of the control group? I also suggest removing the within-group p-value, as the marks already indicate the differences. Replace 'NS' with the actual effect size. Moreover, please refer to other high-quality randomized controlled trial literature to adjust your tables to make them more beautiful and easier for readers to understand.

Please carefully check the grammar and spelling of the article

Experimental design

no comment

Validity of the findings

no comment

Additional comments

no comment

---

## Round 0.3 · Minor Revisions

· Academic Editor

Minor Revisions

Dear Co-Authors:

We are close to finishing the improvements. Please attend to this minor review.

Regards

Dr. Manuel Jiménez

Reviewer 1 ·

Basic reporting

No comments

Experimental design

No comments

Validity of the findings

Line 332: "....the Wilcoxon and Mann-Whitney tests were used to analyze within and between-
333 group differences, respectively."
Dear authors, the Wilcoxon test and the Mann-Whitney U test are the same test! Its correct name should be written: Wilcoxon Mann-Whitney test. This is an error that needs to be corrected!
Line 315: "....315 statistical software (G*Power software, version 3.1.9.4, University of Kiel, Kiel, Germany)." - you still haven't provided references to the statistical tool, you only gave the software version. Tables - their descriptions need to be corrected, because from your description it seems that for you the most important are only the values of the mean and standard deviation.

Additional comments

Please refer to all the suggestions in the earlier review.

·

Basic reporting

no comment

Experimental design

no comment

Validity of the findings

no comment

Additional comments

I suggest to remove this statement in your introduction: “These hypotheses are supported by findings from studies by Bielec et al. (2017) and Abadi et al. (2017)” I suggested it last time, but you did not change it

Line 269: correct squat jump to SJ, please carefully check all abbreviations

Check tables 3-5, control group (n = 15)

---

## Round 0.4 · Minor Revisions

· Academic Editor

Minor Revisions

Dear Authors,

One of the reviewers has requested clarification in the manuscript regarding the application of statistical tests, specifically the Wilcoxon signed-rank test and the Mann-Whitney U test. To address this concern effectively, please note the following points raised by the reviewer, which need to be clearly explained in the text:

1. Terminological Confusion:
• The text mentions both tests in the context of analyzing a single variable (MBT) to compare both “within-group” and “between-group” differences. Suppose it is not made explicit that the Wilcoxon signed-rank test is applied solely for within-group comparisons and the Mann-Whitney U test only for between-group comparisons. In that case, the impression may arise that the authors do not differentiate correctly between these two tests.

2. Ambiguity in Wording:
• The use of terms like “within-group” and “between-group” within the same sentence to describe both tests could imply that the authors consider the tests interchangeable or understand that both can be applied to the same type of analysis. This could lead the reviewer to believe that the authors see the tests as equivalent.

3. Lack of Clarity in Test Distinction:
• If the text does not explicitly clarify why each test was selected for its specific context (Wilcoxon for within-group comparisons and Mann-Whitney for between-group comparisons), the reviewer may interpret that the authors do not fully understand the differences between the two tests and are using them synonymously.

To resolve this reviewer's comment, we ask that you clarify in the text that the Wilcoxon signed-rank test is applied exclusively for changes within the same individuals or group. In contrast, the Mann-Whitney U test is used for comparisons between distinct groups.

As this is now in the “minor revisions” phase, we urge you to address the reviewer’s request thoroughly. Failure to do so may prevent us from accepting your manuscript for publication in PeerJ.

Thank you for your attention to this important detail.

Sincerely,

Dr. Manuel Jiménez
PeerJ Editorial Team

Reviewer 1 ·

Basic reporting

none

Experimental design

none

Validity of the findings

none

Additional comments

Despite suggestions, the authors have not corrected the statistics. It seems that they do not understand it.
This is the same test with a different name.
Line 331 still no correction as well as throughout the text.

·

Basic reporting

no comment

Experimental design

no comment

Validity of the findings

no comment

Additional comments

no comment

---

## Round 0.5 · accepted · Accept

· Academic Editor

Accept

The authors have Appealed. As the Section Editor, I have reviewed the reviewers comments and agree with the authors that the Wilcoxon test and the Mann-Whitney statistical tests are similar but are not the same statistical test. I therefore recommend the amended manuscript for publication.

Thanks, A/Prof M Climstein

· Appeal

Appeal


· · Academic Editor

Reject

First of all, I appreciate your work in improving the manuscript and attending to the reviewer´s comments, but unfortunately, the reviewer has decided to reject your manuscript.

Thank you for your patience and collaboration.

Dr. Manuel Jiménez

Reviewer 1 ·

Basic reporting

none

Experimental design

The authors still do not understand that the Wilxocon test and the Mann-Whitney test are the same test. So it is still an uncorrected error.

Validity of the findings

The authors still do not understand that the Wilxocon test and the Mann-Whitney test are the same test. So it is still an uncorrected error.

Additional comments

The authors still do not understand that the Wilxocon test and the Mann-Whitney test are the same test. So it is still an uncorrected error.